# The Assembly Process of Free-Living and Particle-Attached Bacterial Communities in Shrimp-Rearing Waters: The Overwhelming Influence of Nutrient Factors Relative to Microalgal Inoculation

**DOI:** 10.3390/ani13223484

**Published:** 2023-11-11

**Authors:** Yikai Shi, Xuruo Wang, Huifeng Cai, Jiangdong Ke, Jinyong Zhu, Kaihong Lu, Zhongming Zheng, Wen Yang

**Affiliations:** 1School of Marine Sciences, Ningbo University, No.169 Qixingnan Road, Beilun District, Ningbo 315832, China; sky710746585@163.com (Y.S.); 13780184841@163.com (X.W.); 2211130089@nbu.edu.cn (J.K.); zhujinyong@nbu.edu.cn (J.Z.); lukaihong@nbu.edu.cn (K.L.); zhengzhongming@nbu.edu.cn (Z.Z.); 2Fishery Technical Management Service Station of Yinzhou District, Ningbo 315100, China; huifengcai2023@163.com

**Keywords:** green water, rearing-water bacteria, bacterial lifestyles, community assembly, driving factors

## Abstract

**Simple Summary:**

Inoculating microalgae has been established as having the potential to enhance the microenvironment of shrimp-rearing water, but its relative role in bacterial community assembly in comparison to nutrient enrichment remains largely unexplored. By inoculating two indigenous dominant microalgae, *Nannochloropsis oculata* and *Thalassiosira weissflogii*, into shrimp-rearing waters, we studied the effect of microalgal inoculation and nutrient enrichment on the assembly of particle-attached lifestyles and free-living bacterial communities in rearing water. The key findings include the following: (i) The inoculation of beneficial microalgae contributed to water purification, and the purifying ability of *T. weissflogii* was better than that of *N. oculata*. (ii) The differences between the particle-attached and free-living bacterial communities were significant in terms of composition, representative bacteria, and driving factors, although their dynamic patterns were similar. (iii) Nutrients were vital direct driving factors for bacterial community assembly; however, microalgae indirectly affected the bacterial community via nutrient absorption and nutrient interactions. This work contributes to revealing the assembly mechanism of the bacterial community in rearing waters, which could provide a scientific basis for maintaining a healthy rearing environment and provide a new avenue for optimizing and developing microbial management strategies.

**Abstract:**

The ecological functions of bacterial communities vary between particle-attached (PA) lifestyles and free-living (FL) lifestyles, and separately exploring their community assembly helps to elucidate the microecological mechanisms of shrimp rearing. Microalgal inoculation and nutrient enrichment during shrimp rearing are two important driving factors that affect rearing-water bacterial communities, but their relative contributions to the bacterial community assembly have not been evaluated. Here, we inoculated two microalgae, *Nannochloropsis oculata* and *Thalassiosira weissflogii*, into shrimp-rearing waters to investigate the distinct effects of various environmental factors on PA and FL bacterial communities. Our study showed that the composition and representative bacteria of different microalgal treatments were significantly different between the PA and FL bacterial communities. Regression analyses and Mantel tests revealed that nutrients were vital factors that constrained the diversity, structure, and co-occurrence patterns of both the PA and FL bacterial communities. Partial least squares path modeling (PLS-PM) analysis indicated that microalgae could directly or indirectly affect the PA bacterial community through nutrient interactions. Moreover, a significant interaction was detected between PA and FL bacterial communities. Our study reveals the unequal effects of microalgae and nutrients on bacterial community assembly and helps explore microbial community assembly in shrimp-rearing ecosystems.

## 1. Introduction

Bacteria are ubiquitous and crucial in various terricolous and aquatic environments, as well as in shrimp-rearing-water environments. Bacteria participate in various energy flows and nutrient cycles in shrimp-rearing ecosystems, which play an important role in the stability of rearing ecosystems [1]. Otherwise, large numbers of probiotic and pathogenic bacteria are present in the community, which affect shrimp health and rearing success [2,3]. It has been generally recognized in recent studies that the composition of a bacterial community can be manipulated to favor shrimp health [4]. Revealing the assembly mechanism of the bacterial community in rearing waters provides a scientific basis for manipulating community composition.

Compared with natural environments, the driving factors for shrimp-rearing-water bacterial communities are more complex due to human activities. Inoculation with microalgae, which is a common rearing activity, could promote or restrain the growth of specific bacteria (e.g., *Bacillus* and *Tenacibaculum*) and affect bacterial community composition [5,6,7]. For instance, a study discovered that supplementing diatoms, a type of microalgae, in the rearing medium led to higher weight gain and more efficient feed conversion in *Litopenaeus vannamei* shrimp [8]. Jensen [9] emphasized the significance of microalgae, specifically *Thalassiosira fluviatilis*, in enhancing the weight gain and survival of *Farfantepenaeus paulensis* shrimp in nursery tanks. Additionally, the introduction of microalgae, particularly *Scenedesmus obliquus*, improved fish survival in an integrated shrimp and fish culture system using biofloc technology [10]. These findings strongly suggest that microalgae inoculation can significantly enhance the performance and sustainability of shrimp-rearing systems. Furthermore, nutrient fluctuations caused by the decomposition of residual feed and shrimp metabolites and the absorption of microalgae are also major constraining factors for bacterial community composition [11,12,13]. However, few studies have considered the relative importance of microalgae and nutrient factors in the assembly of rearing-water bacterial communities.

Traditionally, all bacteria living in water are studied as a whole community. However, bacteria of different lifestyles, mainly particle-attached (PA) and free-living (FL) bacteria, exhibit significant differences in terms of their composition and functions [14]. For example, PA bacteria are probably effective in converting organic matter [15] and assisting microalgal functions [16], whereas FL bacteria might be sensitive to nutrient factors due to their smaller size [14]. Shrimp-rearing-water ecosystems harbor various microhabitats for PA and FL bacteria to form communities, but the distinction between PA and FL bacterial community compositions and assembly processes has not been explored thus far.

Our previous studies separately demonstrated the influence of microalgae [17] and nutrients [13] on the overall rearing-water bacterial community, but the relative contributions of these two factors have not been compared. To this end, we introduced *Nannochloropsis* oculata and *Thalassiosira weissflogii* into *Litopenaeus vannamei* rearing waters to study (1) the temporal patterns of nutrient factors, microalgae, and bacterial communities in rearing waters and (2) the effects of microalgae and nutrient factors on PA and FL bacterial community composition, diversity, assembly processes, and co-occurrence patterns.

## 2. Materials and Methods

### 2.1. Microalgae Culture

*N. oculata* and *T. weissflogii* were both obtained from the Marine Biotechnology Laboratory of Ningbo University, China. N. oculate was cultivated in an NMB3 medium with seawater filtered through 0.45 μm cellulose acetate membranes and sterilized via autoclaving [17]. For the culture medium of *T. weissflogii*, Na_2_SiO_3_ (2 mg/L) was added to the NMB3 medium. Five-liter glass conical flasks sterilized via autoclaving were employed for microalgal primary incubation at a light intensity of 100 mmol photons/(m^2^·s) and a temperature of 27 °C. Subsequently, the microalgae in the flasks were inoculated into 10 L plastic cylindrical photoreactors sterilized with hypochlorous acid (HOCl) for microalgal secondary productive incubation. These microalgae in the exponential growth phase were used for subsequent experiments after the concentration of *N. oculata* reached 10^7^ cells/mL and the concentration of *T. weissflogii* reached 10^6^ cells/mL.

### 2.2. Experimental Design

The shrimp *Litopenaeus vannamei* and rearing water for the experiment were obtained from a shrimp pond in Xiangshan Lanshang Marine Technology Co., Ltd., Ningbo, China (29°28′ N, 118°6′ E) and were transferred to the production base of Ningbo University, Ningbo, China (29°46′ N, 121°57′ E), where the experiment was conducted. Approximately two thousand shrimp with a body length of about 5 cm and good activity were selected and acclimated for 10 days. The native microalgal communities of the rearing water in the shrimp pond were dominated by *Oocystis borgei* and *Cyclotella* spp., with a biomass of about 25 mg/L. After being transferred to the production base, the rearing water was precipitated for 10 days and prefiltered sequentially through a 100 μm and 1 μm disinfected nylon mesh to remove large particles and native microalgae. Nine 500 L polyethylene fiber tanks were randomly divided into three groups: 450 L filtered water was transferred to a tank for the control group (Group C); Group N consisted of 450 L of filtered water and 4.5 L of *N. oculata* suspension (the *N. oculata* concentration was approximately 3 × 10^5^ cells/mL); and Group T was 450 L of filtered water and 4.5 L of *T. weissflogii* suspension (the *T. weissflogii* concentration was approximately 8 × 10^3^ cells/mL). Considering the difference in biological volume between *N. oculata* and *T. weissflogii*, the initial biomass of microalgae was set at about 25 mg/L based on the microalgal biomass of the shrimp pond. Each group had 3 replicates. Each tank was stocked with 150 shrimp. Electric aerators were used to aerate all tanks. Commercial feed (fish meal, 38%; yeast powder, 4%; soybean lecithin, 18%; peanut powder, 6%; shrimp shell powder, 10%; wheat gluten powder, 10.5%; vegetable oil, 1.5%; Ca(H_2_PO_4_)_2_, 3%; and premixture, 4%) was used to feed the shrimp twice daily (at 8:00 a.m. and 8:00 p.m.). The experiment started on 25 October 2020 and ended on 3 November 2020. The tanks were operated in zero water exchange mode throughout the entire experimental period.

### 2.3. Environmental and Bacterial Sample Collection

Duplicate 500 mL water samples were collected in each tank on days 1, 4, 7, and 10 after the initiation of the experiment and kept in sterile polyethylene bottles. In total, 72 (3 groups × 3 replicates × 4 time points × 2 duplicates) water samples were collected. The samples were stored in the dark at 4 °C and transported to the laboratory for further processing.

Within 3 h after sampling, each water sample was first filter-sterilized through a 3 µm pore size polycarbonate membrane (47 mm diameter, Millipore, Boston, MA, USA) to collect particle-attached bacteria. The 3 µm filtrate water was then filter-sterilized through a 0.22 µm pore size polycarbonate membrane (47 mm diameter, Millipore, Boston, MA, USA) to collect free-living bacteria. Two membranes with the same pore size and from the same tank were placed in one sterilization tube and stored at −80 °C as one bacterial sample. In total, 36 (3 groups × 3 replicates × 4 time points) PA bacterial samples and 36 (3 groups × 3 replicates × 4 time points) FL bacterial samples were collected.

The 0.22 µm filtered water was kept at 4 °C in 10 mL sterilization tubes and analyzed with an automated spectrophotometer (Smart-Chem 450 Discrete Analyzer, Westco Scientific Instruments, Brookfield, WI, USA) to determine the concentrations of ammonium (NH_4_^+^), nitrite (NO_3_^−^), nitrate (NO_2_^−^), and orthophosphate (PO_4_^3−^) within 48 h.

Approximately 150 mL amounts of the water samples were transferred to a glass bottle and fixed with Lugol’s solution for storage. The microalgae were identified and counted in sedimentation chambers (Hydro-Bios Apparatebau GmbH, Kiel, Germany) with an inverted microscope (CK2, Olympus Corporation, Tokyo, Japan) according to “Flora Algarum Marinarum sinicarum” [18]. Phytoplankton biomass was calculated via geometric approximations using a computerized counting program (OptiCount, https://science.do-mix.de/software_opticount.php, accessed on 3 November 2023).

### 2.4. Bacterial Illumina HiSeq Sequencing and Data Bioinformatic Analyses

Bacterial DNA was extracted from 72 (36 PA bacteria and 36 FL bacteria) bacterial samples via a MinkaGene Water DNA kit (Guangdong Magigene Biotechnology Co., Ltd., Guangzhou, China), and its concentration and purity were measured with a NanoDrop One spectrophotometer (Thermo Fisher Scientific, Waltham, MA, USA). The Invitrogen (Invitrogen, Carlsbad, CA, USA) synthesized primers 515F (5′-GTGCCAGCMGCCGCGGTAA-3′) and 806R (5′-GGACTACNNGGGTATCTAAT-3′) were used with barcodes to amplify the V4 hypervariable region of the bacterial 16S rRNA gene [19], with reactions involving 25 μL of 2× Premix Taq (Takara Biotechnology, Dalian Co., Ltd., Dalian, China), 1 μL of each primer (10 nM), and 3 μL of DNA template (20 ng/μL) in a volume of 50 μL and thermocycling conditions consisting of predenaturation at 94 °C for 5 min; 30 cycles of denaturation at 94 °C for 30 s; annealing at 52 °C for 30 s; extension at 72 °C for 30 s; and a final elongation step at 72 °C for 10 min. Next, three equimolar PCR amplification products were purified and combined using the NEBNext^®^ Ultra™ DNA Library Prep Kit for Illumina^®^ (New England Biolabs, Ipswich, MA, USA) with a sequencing library. Finally, the library was sequenced using the Illumina HiSeq 2500 platform (Guangdong Magigene Biotechnology Co., Ltd., Guangzhou, China) to generate 250 bp paired-end reads.

The sequenced paired-end reads were deposited in the NCBI Sequence Read Archive with the BioProject number PRJNA881623 and the accession number SRP397855 (https://www.ncbi.nlm.nih.gov/bioproject/PRJNA881623, 19 September 2023) and processed using USEARCH V.11 [20]. First, the paired-end reads were merged and denoised (unoise_ alpha = 2 and minsize = 4 as per default settings) using the UNOISE3 pipeline [21]. Then, the filtered sequences were clustered into zero-radius operational taxonomic units (ZOTUs). The SILVA database (release v138) for bacteria was used to assign the representative sequences for each ZOTU at 99% similarity using the RDP classifier.

### 2.5. Data Statistical Analyses

All statistical analyses were carried out in an R software environment (version 4.1.3) and visualized using the “ggplot2” package. One-way analysis of variance (ANOVA) was employed with the function aov () from the stats package to test the significance of the nutrient factors among the treatment groups. Principal component analysis (PCA) was adopted via the function rda () in the “vegan” package to show the correlation between microalgal and nutrient factors. Spearman’s correlation was performed between the principal components and environmental factors via the cor () function from the “vegan” package. Analysis of similarity (ANOSIM) was performed using the anosim () function in the “vegan” package to confirm a statistically significant difference between two groups. The “Venn” package was used to display the number of ZOTUs in each group.

Principal component analysis (PCoA) was calculated on the basis of the Bray–Curtis distance of whole (both PA and FL) bacterial communities and performed using the cmdscale () function from the “ape” package. Constrained principal component analysis (CPCoA) based on Bray–Curtis metric dissimilarities in the PA and FL bacterial communities was performed by the capscale () function in the “vegan” package and the capscale () function in the “vegan” package. A quantitative assessment of the effects of rearing time and different microalgal treatments on variations in the whole, PA, and FL bacterial communities based on nonparametric multivariate analysis of variance (PERMANOVA) was performed using the adonis () function from the “vegan” package.

Canonical correspondence analysis (CCA) and the Mantel test were performed on the physicochemical factors and bacterial communities to relate bacterial community succession to environmental factors. The analyses above were conducted using the “vegan” package. Ternary plots and heatmaps were employed to elucidate the relative relationships and distributions of dominant species among the three treatments using the “ggtern” and “edgeR” packages. Partial least squares path modeling (PLS-PM) was performed by using the plspm () function in the “plspm” package.

### 2.6. Co-Occurrence Network Analysis

Some low-abundance species were removed from the bacterial abundance, and then, relative abundance counts per million (CPM) were determined by taking the intersection of the significant between-group difference ZOTUs based on the indicated species and edgeR. Pairwise Spearman’s correlation calculations were based on the TMM standardized ZOTU table. After P value correction, the relationship with Spearman’s rho > 0.7 and *p* value < 0.001 was selected as the selected co-occurrence network. The correlation between the abundance of each module and the total variation in the measured environmental factors was calculated.

## 3. Results

### 3.1. Variations in Microalgal and Nutrient Factors

There were no significant differences observed in shrimp length and weight. Distinct nutrient levels were observed among the different microalgal treatments, although they all increased as the experiment proceeded (Figure 1A–D). Overall, the concentrations of PO_4_^3−^ in Group T were lower than those in Group N and Group C, with significant differences at days 4 and days 7 (Figure 1D). Furthermore, the concentrations of NO_3_^−^ in the treatment groups (Group N and Group T) were higher than those in the control group, with significant differences at days 4 and days 7 (Figure 1C). After the inoculation of *N. oculata* and *T. weissflogii*, we found that the microalgal biomass of Group N and Group T increased and remained essentially stable for 4–10 days, with *N. oculata* at 157.79 ± 13.96 mg/L and *T. weissflogii* at 443.89 ± 47.10 mg/L (Figure 1E). Principal component analysis showed that PC1 factors were significantly and positively correlated with the concentrations of PO_4_^3−^, NO_3_^−^, NO_2_^−^-, and NH_4_^+^ (Figure 1F, Appendix A), while the biomasses of *N. oculata* and *T. weissflogii* exhibited significant correlations with PC2 (Figure 1F, Appendix A). Therefore, PC1 and PC2 could be used to represent the overall variations in nutrient and microalgal factors, respectively.

### 3.2. Dynamics of Bacterial Community Composition and Diversity

In total, 9,061,540 high-quality sequences were obtained from 72 samples, which generated 7872 bacterial ZOTUs via downstream analysis (Appendix A). These ZOTUs were mainly assigned to Alphaproteobacteria (75.41% ± 21.52% of the PA bacterial community; 64.82% ± 15.55% of the FL bacterial community), Gammaproteobacteria (11.58% ± 17.08% of the PA bacterial community; 15.95% ± 15.14% of the FL bacterial community), and Bacteroidetes (6.21% ± 5.73% of the PA bacterial community; 11.17% ± 6.98% of the FL bacterial community) (Appendix A), but significant differences in taxonomic composition were detected between lifestyles [ANOSIM, R^2^ = 0.062, *p* = 0.02], groups [ANOSIM, R^2^ = 0.113, *p* = 0.003], and along rearing times [ANOSIM, R^2^ = 0.682, *p* = 0.001]. For the α-diversity of the bacterial communities, significant differences were not found between PA and FL but were found among groups (Appendix A).

Initially, the bacterial communities of the three groups were similar based on MRPP, ANOSIM, and Adnois (Appendix A). PCoA based on the Bray–Curtis dissimilarities of all samples showed that both the PA and FL bacterial communities varied over time (Figure 2A), but their temporal patterns were different [ANOSIM, R^2^ = 0.682, *p* = 0.001]. PERMANOVA showed that the variations in the overall (both PA and FL), PA, and FL bacterial communities were mainly affected by rearing time (explaining 22.13%, 23.57%, and 25.27%, respectively), as well as microalgae inoculation (treatment) and their interaction (Table 1). Similarly, in the CPCoA for the PA (Figure 2B) and FL (Figure 2C) bacterial communities, the clusters that were in line with groups emphasized the effect of treatment. Regression analyses revealed the negative relationship between the PCo 1 of PCoA and PC1 (Appendix A), suggesting the effect of nutrient factors on bacterial community structure.

### 3.3. The Representative Bacterial Genera of Different Microalgal Treatments and Their Relationships with the Environment

Indicator species analyses and likelihood ratio tests were employed to identify the representative genera of different microalgal treatments in the PA and FL bacterial communities. A total of 23 and 14 genera were identified in the PA and FL bacterial communities, respectively (Figure 3). Among them, three genera (*Ruegeria*, *Tropicibacter,* and *Haliea*) in Group C, three genera (*Oceanicaulis*, *Sulfitobacter,* and *Yangia*) in Group N, and two genera (*Hyphomonas* and *Marivita*) in Group T were the representative genera for both the PA and FL bacterial communities (Figure 3). Moreover, in the PA bacterial communities, *N. oculata* specifically enriched *Celeribacter*, *Ponticoccus*, *Algoriphagus,* and *Thalassococcus* (Figure 3A), while *T. weissflogii* specifically enriched *Paracocccus* and *Henriciella* (Figure 3A). For Group C, nine genera, including *Polaribacter*, *Marinicella,* and *Thioclava*, could specifically represent PA bacterial communities (Figure 3A), while six genera, including *Synechococcus*, *Lutimaribacter,* and *Balneola*, could specifically represent PA bacterial communities (Figure 3B). Spearman’s correlation coefficients were calculated between the representative bacteria and each environmental factor. Almost every genus was significantly (*p* < 0.05) correlated with at least one environmental factor (Figure 3C,D).

### 3.4. The Influence of Environmental Factors on Bacterial Communities

Overall, the PA and FL bacterial communities were significantly associated with nutrient factors (PC1) based on the Mantel test, but no significant associations were identified with microalgae (PC2) (Table 2). CCA also highlighted the influence of nutrient factors on bacterial community composition (Appendix A). However, PLS-PM analysis showed that microalgae could directly affect the PA bacterial community and nutrient factors PO_4_^3−^ and NH_4_^+^ (Figure 4). Furthermore, the affected PO_4_^3−^ and NH_4_^+^ had significant positive and negative influences on NO_2_^−^, respectively (Figure 4). Moreover, there were significant interactions between the PA and FL bacterial communities (Figure 4).

### 3.5. The Co-Occurrence Patterns of Bacterial Communities and Their Relationships with the Environment

A co-occurrence network containing 7872 ZOTUs and 1364 significant positive correlations was constructed, and four modules (M1, M2, M3, and M5) that contained relatively high proportions of the representative bacteria from different treatments (Figure 5A) were found. Specifically, M1 was made up of bacteria that were significantly represented in the microalgal treatment groups, while M2 was made up of bacteria that were significantly represented in the control group (Figure 5B). Moreover, representative PA bacteria were mostly associated with M3, while representative FL bacteria were mostly associated with M5 (Figure 5B). Regression analyses of bacterial co-occurrence patterns and environmental factors revealed that all the modules were significantly correlated with PC1, but only M3 was significantly correlated with PC2 (Table 3).

## 4. Discussion

### 4.1. The Distinct Patterns of PA and FL Bacterial Communities in Shrimp-Rearing Waters

Clarifying the assembly patterns of the bacterial community in shrimp-rearing waters is of great significance for improving aquaculture technologies. Our study innovatively explored the bacterial community assembly of different lifestyles that are artificially categorized into two in aquatic environments: PA and FL [22,23]. Previous studies have speculated that PA and FL bacteria occupy different ecological niches, although they both live in the same environment. Additionally, some studies have demonstrated that the compositions, assembly processes, and ecological functions of PA and FL bacterial communities are significantly different [23,24]. However, others have argued that these bacteria are essentially the same and play slightly different biogeochemical roles due to their possible lifestyle transitions [25,26].

Thus, the similarities or dissimilarities between PA and FL bacterial communities might depend on environmental factors. For shrimp-rearing waters, we found that the compositions of the PA and FL bacterial communities were significantly different (Table 1), but their dynamic patterns were similar (Figure 2A). Although a large amount of overlap was identified in the representative genera of the PA and FL bacterial communities, there were still some representative genera that could live only freely or while attached (Figure 3), such as *Marinicella* and *Algoriphagus*, which are consistently attached to microalgae as phycosphere-associated bacteria (Figure 3C) [27,28], and *Peredibacter*, which settle on free-living life (Figure 3D) [29,30], leading to distinct patterns in the PA and FL bacterial communities. It is worth noting that the representative bacteria in the PA communities were more abundant than those in the FL communities (Figure 3C,D). One possible reason for this phenomenon is that the variations in the FL bacterial communities were transferred from those in the PA bacterial communities in view of the lifestyle transition between free-living and particle-attached states [26]. Our results showed that the microalgal and nutrient factors in shrimp-rearing waters first directly affected PA bacterial communities (Figure 4); thus, more representative PA bacteria responded to environmental factors and provided a variable source for FL bacterial communities (Figure 4). Other studies in mesocosms and fields have also revealed that PA bacteria are important sources for their FL counterparts [25,31]. This is also why representative PA and FL bacteria had a large amount of overlap. Another possible reason for the more abundant representative bacteria in the PA communities is that particles have higher substrate availabilities (niches) in their microenvironments than free-living habitats [14,32], especially in shrimp-rearing waters. The discrepancies in nonliving particles (i.e., residual feeds, feces, and shrimp residues) and living particles (i.e., microalgae) under different treatments created diverse microhabitats for bacteria [33,34], which led to the abundant representative PA bacteria in different treatments.

### 4.2. The Unequal Effects of Microalgae and Nutrient Factors on Bacterial Community Assembly

Shrimp-rearing practices could result in an overload in nutrient inputs and in metabolic wastes that are retained and degraded in a water ecosystem, which could cause gradual water eutrophication or even water quality deterioration [13,35,36]. This study examined a zero-water exchange rearing system to reduce pollution in the surrounding sea areas and achieve environmental friendliness; even so, the current rearing management strategies could only alleviate the increasing trend in nutrients rather than eliminate them (Figure 1) [17,37,38]. In our study, the microalgal biomass was maintained and exploded after inoculation, which could be attributed to the continuous enrichment of nutrients in the rearing water (Figure 1). However, the short-term effect of microalgal absorption was limited, judging from the subsequent variable trend in nutrient factors (Figure 1). Therefore, a gradual enrichment in nutrients is an important factor that constrains bacterial communities in shrimp-rearing waters. This phenomenon aligns with our prior research findings [9,10,17]. Moreover, the water discharged from a shrimp-rearing system can undergo treatment through external aquaculture wastewater treatment systems [39]. These systems utilize bivalves and large seaweeds to enhance water purification, thus averting pollution in the neighboring marine environment [40,41,42].

Additionally, our results showed that nutrient factors could significantly affect the diversity, composition, representative bacteria, and co-occurrence patterns of rearing-water bacterial communities (Figure 2 and Figure 3; Table 2, Table 3 and Appendix A), which echoes our previous findings [12,13,17] and those of other researchers [43,44]. The PLS-PM analysis showed that nitrite nitrogen was a crucial factor that affected the bacterial community, constraining the overall bacterial community through a direct correlation with the PA community and an indirect correlation with the FL community (Figure 4). Similarly, Acinas et al. (2021) indicated that nitrate pathways enriched in PA bacteria play significant roles in deep-ocean microbial communities [45]. It is worth noting that the variation in nitrite nitrogen was determined by other nutrient factors (Figure 4), which emphasized the interactions among nutrient factors [46,47] and their effects on bacterial community assembly. In other words, the relationships that were detected via traditional statistical analyses between individual nutrient factors and the bacterial community might be indirectly derived from the interactions of nutrient factors, which provides a new perspective for unravelling the mechanisms of bacterial community assembly.

As a green water technology strategy, the inoculation of certain beneficial microalgae into rearing waters could improve rearing environments and stabilize rearing systems [15,17,48]. 

Our results found that the inoculation of beneficial microalgae contributed to water purification, and the purifying ability of *T. weissflogii* was better than that of *N. oculata* (Figure 1). Indeed, other studies have also confirmed the purifying capabilities of *Thalassiosira* [49,50]. It is generally believed that microalgae affect bacterial communities through two pathways: direct effects through microalgae–bacteria metabolic exchanges [51] and indirect effects mediated by nutrient factors [52,53]. However, our results showed that, over a short period of time, the direct effects of microalgae on bacterial community assembly were less prominent (Figure 2 and Figure 3; Table 2 and Table 3), and the effects of microalgae were mainly realized although nutrient factors (Figure 4). A possible reason for this phenomenon is that in a resource-rich environment, such as our rearing waters, microalgae might prefer to reproduce by absorbing nutrients rather than by attracting bacteria to establish their own phycosphere. However, although the effects of microalgae on bacteria at the community level were not significant, different microalgal treatments had divergent effects on certain bacteria. For rearing waters inoculated with microalgae, microalgal symbiotic bacteria, such as *Celeribacter* [54] and *Ponticoccus* [55] for *N. oculata* and *Marivita* [28] and *Hyphomonas* [56] for *T. weissflogii*, were specifically promoted (Figure 3). For rearing waters that did not contain microalgae, the dominant bacteria were mainly bacteria that decomposed organic matter (i.e., *Polaribacter*) [57] or were engaged in bacterial symbiosis with dinoflagellates (i.e., *Tropicibacter*) [58], or they were autotrophic bacteria that replaced microalgae as the producers in the water ecosystem (i.e., *Thioclava*) [59] (Figure 3). By incorporating these results, we can infer that both the presence/absence of microalgae and the species of microalgae can determine bacterial community assembly in shrimp-rearing waters. Therefore, screening suitable microalgal species according to the conditions of the rearing environment is a first step in applying green water technology.

## 5. Conclusions

This study delved into the distinct effects of microalgal inoculation and nutrient factors on shrimp-rearing-water bacterial communities. The findings revealed that microalgae inoculation treatment could mitigate nutrient enrichment in shrimp-rearing environments and influence the assembly of bacterial communities, although these effects vary depending on the specific microalgae species used. Additionally, our study highlighted significant disparities in composition, representative bacteria, and driving factors between the PA and FL bacterial communities, even though their dynamic patterns exhibited similarities. Notably, nutrients emerged as crucial direct driving forces behind bacterial community assembly. Intriguingly, microalgae influenced the bacterial community indirectly by influencing nutrient absorption and interactions. In essence, our findings shed light on the unequal impacts of microalgae and nutrient factors on the assembly of bacterial communities, particularly in bacteria with distinct lifestyles. This understanding is pivotal for advancing our comprehension of microbial community assembly in shrimp-rearing ecosystems. Furthermore, it provides valuable insights for the development of sustainable green water technologies. However, it is important to acknowledge the potential limitations of our approach. For instance, in future studies, incorporating negative controls to account for bacterial contamination from both shrimp and microalgae suspensions will be essential to enhance the robustness of our findings.

## Figures and Tables

**Figure 1 animals-13-03484-f001:**
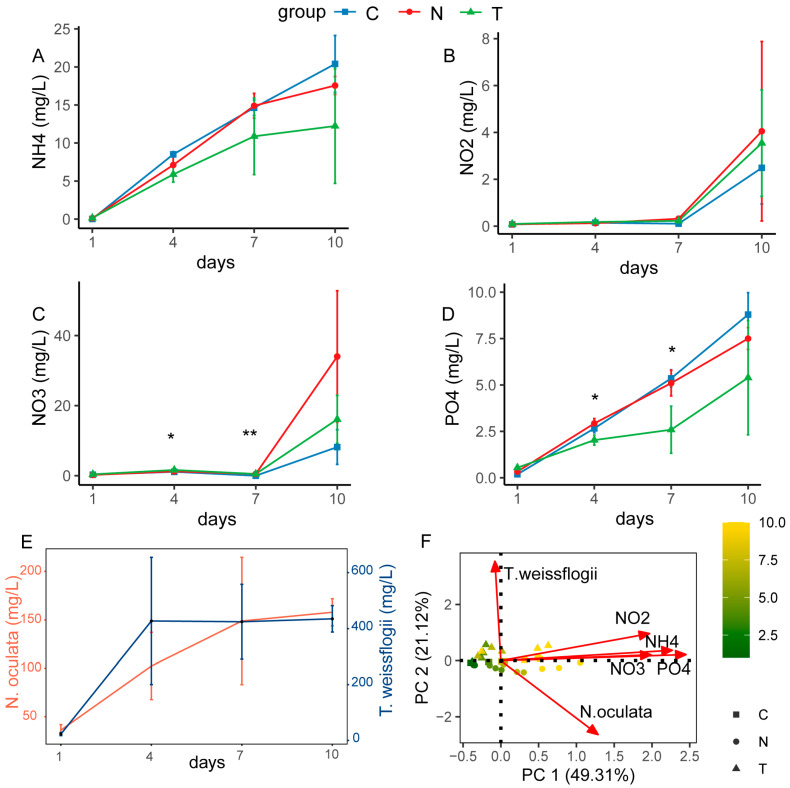
Variations in microalgal and nutrient factors of rearing water during experiment period in Group N (filtered rearing water inoculated with *N. oculata*), Group T (filtered rearing water inoculated with *T. weissflogii*), and Group C (control, filtered rearing water). (**A**–**D**) Changes in nutrient factors. (**E**) Changes in *N. oculata* and *T. weissflogii* biomass. (**F**) Principal components analysis (PCA) of microalgal and nutrient factors. “*” represents a significant difference among groups on the same day (* *p* < 0.05, one-way ANOVA. ** *p* < 0.01). Error bars represent standard errors of the means (n = 3).

**Figure 2 animals-13-03484-f002:**
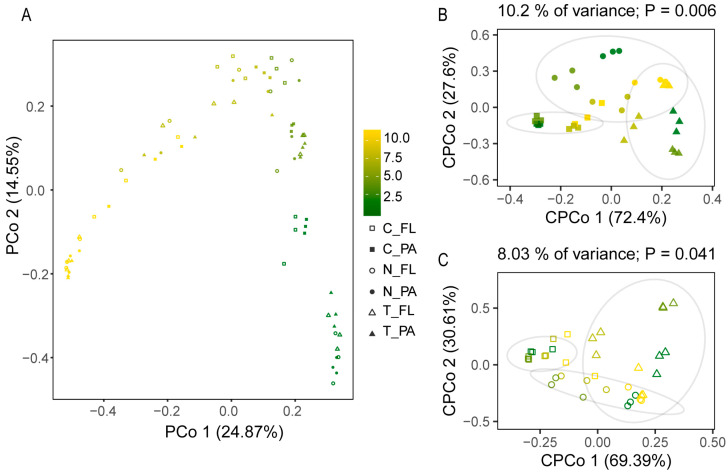
Dynamics of bacterial communities’ structure in three groups. (**A**) Principal coordinates analysis (PCoA) based on the Bray–Curtis dissimilarities of whole (both PA and FL) bacterial communities. Constrained principal coordinates analysis (CPCoA) based on the Bray–Curtis dissimilarities of PA (**B**) and FL (**C**) bacterial communities.

**Figure 3 animals-13-03484-f003:**
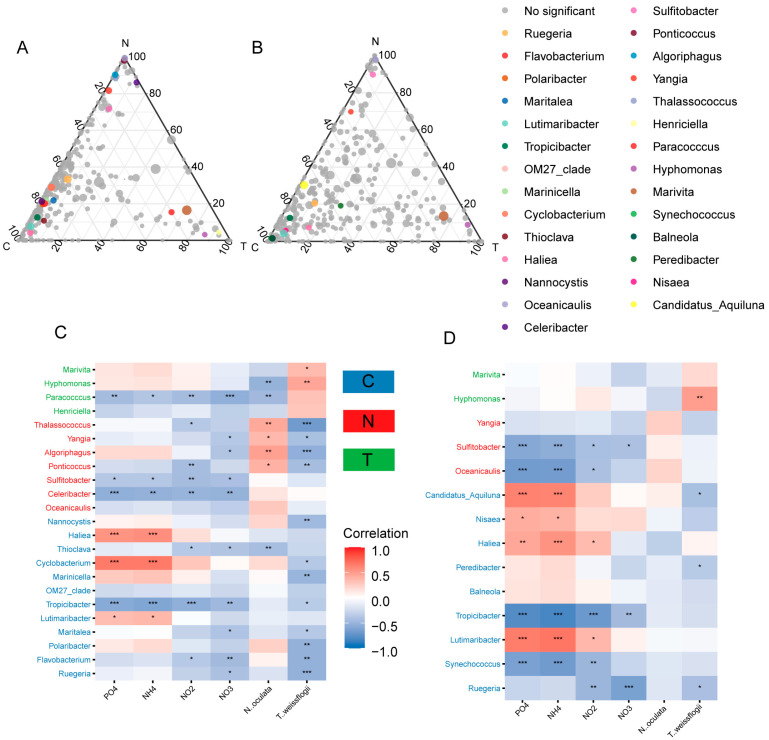
Representative bacterial genera and their relationships with environmental factors. Ternary plots illustrating the distributions of representative bacterial genera in PA (**A**) and FL (**B**) bacterial communities. Spearman’s correlation between environmental factors and representative bacterial genera in PA (**C**) and FL (**D**) bacterial communities. * *p* < 0.05; ** *p* < 0.01; *** *p* < 0.001.

**Figure 4 animals-13-03484-f004:**
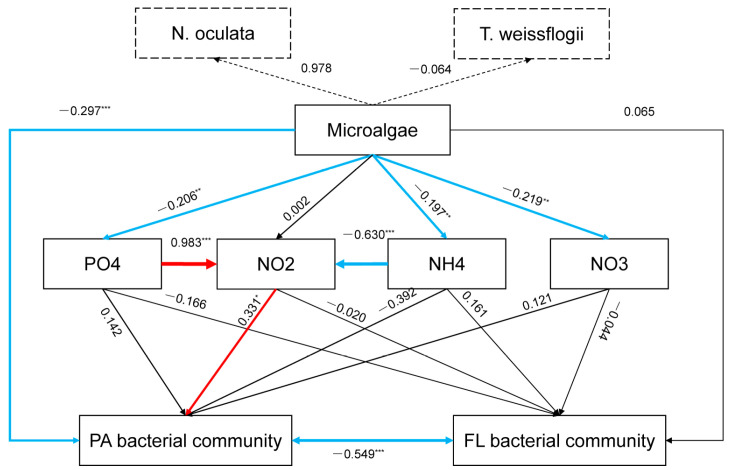
PLS-PM analysis showing the direct and indirect effects of macroalgae and nutrient factors on bacterial communities. The red and blue lines indicate significant (*p* < 0.05) positive and negative associations, respectively. The black lines indicate nonsignificant (*p* > 0.05) associations. * *p* < 0.05; ** *p* < 0.01; *** *p* < 0.001.

**Figure 5 animals-13-03484-f005:**
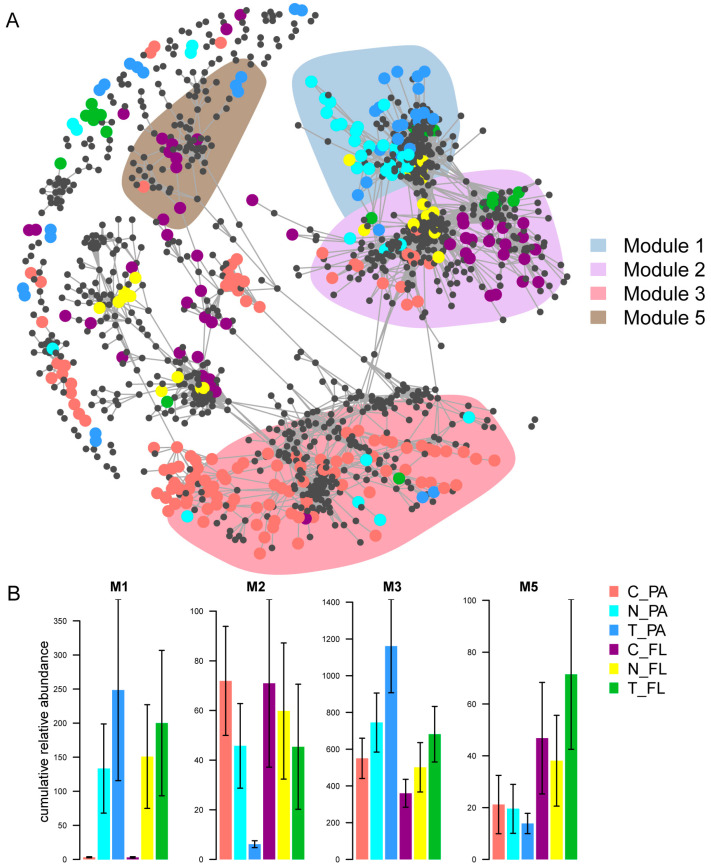
Co-occurrence patterns of whole bacterial communities in shrimp-rearing water. (**A**) Co-occurrence network of whole bacterial communities and representative bacteria of different treatments and different lifestyles. (**B**) Cumulative relative abundances of representative bacteria of different treatments and different lifestyles. Different colors of nodes represent representative bacteria in different treatments and different lifestyles. Shaded areas represent the network modules containing representative bacteria.

**Table 1 animals-13-03484-t001:** Quantitative assessment of the effects of rearing time and different microalgal treatments on variations in whole, PA, and FL bacterial community structures based on nonparametric multivariate analysis of variance (PERMANOVA).

Factors	Whole	PA	FL
Treat	0.086 ***	0.120 ***	0.089 **
Time	0.221 ***	0.236 ***	0.253 ***
Lifestyles	0.083 ***	0.109 ***	0.083 **

Note: R^2^ represents the contribution of individual variables and interactions between variables to drive changes in the structure of the bacterioplankton community. ** *p* < 0.01; *** *p* < 0.001.

**Table 2 animals-13-03484-t002:** Mantel test comparison between bacterial community and environment factors.

	Whole	PA	FL
PC	0.226 ***	0.221 ***	0.257 ***
PC1	0.289 ***	0.298 ***	0.360 ***
PC2	0.031	0.012	0.009

Note: R^2^ represents the correlation coefficients and the given correlations between bacterial community and PCA axes. *** *p* < 0.001.

**Table 3 animals-13-03484-t003:** Regression analyses of co-occurrence network modules and environment. * *p* < 0.05; *** *p* < 0.001.

	TM1	TM2	TM3	TM5
PC1	−3.827 ***	5.102 ***	4.305 ***	2.163 *
PC2	1.316	−1.971	3.781 ***	1.221

## Data Availability

Data are contained within the article and Appendix A. The sequenced paired-end reads were deposited in the NCBI Sequence Read Archive with the BioProject number PRJNA881623 and the accession number SRP397855 (https://www.ncbi.nlm.nih.gov/bioproject/PRJNA881623, 19 September 2023).

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
