# Peer review of "The Assembly Process of Free-Living and Particle-Attached Bacterial Communities in Shrimp-Rearing Waters: The Overwhelming Influence of Nutrient Factors Relative to Microalgal Inoculation"

_animals, 2023, doi:10.3390/ani13223484_

Round 1

Reviewer 1 Report

Comments and Suggestions for Authors

It is a valuable MS. However I made some comments and questions to consider on the attached MS. Introduction should be enlarged and M&M amended a bit.

My reference search by Elicit has resulted:

The papers collectively suggest that the use of microalgae inoculation in shrimp-rearing systems can have positive effects. Cesar 2012 and Godoy 2011 found that supplementing diatoms, a type of microalgae, in the rearing medium resulted in higher weight gain and more efficient feed conversion in Litopenaeus vannamei shrimp. Jensen 2006 also highlighted the importance of microalgae, specifically Thalassiosira fluviatilis, in improving weight gain and survival of Farfantepenaeus paulensis shrimp in nursery tanks. Additionally, Silva 2022 demonstrated that the addition of microalgae, specifically Scenedesmus obliquus, improved fish survival in an integrated shrimp and fish culture system using biofloc technology. These findings suggest that microalgae inoculation can enhance the performance and sustainability of shrimp-rearing systems.

Author Response

Response to Reviewer 1 Comments

We are very grateful for your affirmation of our work. We greatly appreciate your comments and the thorough revision that greatly contributed to the improvement of our manuscript. Please find the detailed responses below. Red color indicated the corresponding revisions in the re-submitted files.

Comments 1: Introduction: This part seems to be very short. I miss a few references that should have been mentioned helping to give an ampler view on the area. Mentioning green water and biofloc technologies, their importance and popularity etc. See my attached word file of searching results.

Responses 1: We appreciate your comments, and as per your suggestion, the introduction has been expanded accordingly. The revision can be seen in the revised manuscript, Line 59-67.

Comments 2: critical?

Responses 2: Thank you for pointing out this error. The word "critical" has been changed to "crucial", as can been seen in the revised manuscript, Line 46.

Comments 3: Line 101-103. This should be justified. If you had any intention to make some kind of a model of practical shrimp production, why the filtration of even the control tanks' water was needed? If this wasn’t a goal, please justify also. On the other hand, how the efficiency of this process was checked?

Responses 3: We are deeply appreciative of your suggestion. Given the presence of large particles such as leftover feed and feces in the culture water sourced from the shrimp pond, which could lead to differing initial environmental conditions in each tank, we employed a 100-micron mesh filter to eliminate these potentially disruptive particles. Due to the 100-micron pore size primarily capturing larger organic particles, the filtration rate is remarkably swift, and the culture water is filtered almost instantly.

Comments 4: Line 115. Daily ration used to be given as % of the animals biomass. On the other hand, the exact amount of this input may be an important detail.

Responses 4: We acknowledge the importance of exact amount of daily feeding ration as crucial data. Regrettably, we records only include the initial and final shrimp weights throughout the experiment, lacking the daily feeding amounts for each tank. We sincerely apologize for this oversight and assure you that we will give strong consideration to this aspect in our future experimental designs

Comments 5: Line 119. 10 days seem to be rather a short period for an experiment like this. Were ther any previous experiencies that justify this methodology?

Responses 5: We greatly appreciate your suggestion. In our previous research, we have demonstrated the positive impact of microalgae on aquaculture water and shrimp health (Yang et al., 2020; Ding et al. 2021). In this experiment, our focus is on studying the influence of different microalgae species on the bacterial communities in shrimp rearing water. Due to the small size and rapid reproduction of bacteria, the bacterial communities can undergo succession within 10 days under the influence of microalgae. Therefore, we have chosen a 10-day experimental duration for this study.

Yang W, Zheng Z, Lu K, Zheng C, Du Y, Wang J, Zhu J*. Manipulating the phytoplankton community has the potential to create a stable bacterioplankton community in a shrimp rearing environment. Aquaculture, 2020, 520: 734789.

Ding Y, Chen N, Ke J, Qi Z, Chen W, Sun S, Zheng Z, Xu J, Yang W*. Response of the rearing water bacterial community to the beneficial microalga Nannochloropsis oculata cocultured with Pacific white shrimp (Litopenaeus vannamei). Aquaculture, 2021, 542: 736895.

Comments 6: 3.1. And what happened to the shrimp? Survival, growth? Isn't that interesting?

Responses 6: We apologize for the initial omission of these crucial data. The weight and length of the shrimp were measured before stocking and at the end of the experiment. However, as no significant differences were observed among the three treatments, we chose not to include the presentation of this data. In response to your feedback, we have added corresponding descriptions in the revised manuscript, Line 215.

Comments 7: Table 2 Note. R2 is called coefficient of determination. However, it may represent the given correlation.

Responses 7: Thank you for your suggestion. The note has been revised and can be found in the revised manuscript, Line 308.

Reviewer 2 Report

Comments and Suggestions for Authors

This article is devoted to the study of the assembly of free-living and particle-attached bacterial communities in the waters where shrimp are grown, the influence of nutrient media, etc. The article is written in good and understandable language. The relevance of this work, in general, is beyond doubt. The study of bacterial communities plays an important role in understanding some global processes occurring in living nature. In terms of the level of data, volume and subject matter, this work meets the requirements of the Journal. There are some points for improvement:

1. It is advisable to expand the introduction.

2. It is desirable to more clearly define the purpose of this study.

3. It is advisable to compare all experimental results in more detail with data from the literature.

4. Was the experimental methodology developed by the authors or modified? What did the authors rely on in selecting the study parameters?

5. It is advisable to expand the conclusions

Author Response

Response to Reviewer 2 Comments

We are very grateful for your affirmation of our work. We greatly appreciate your comments and the thorough revision that greatly contributed to the improvement of our manuscript. Please find the detailed responses below. Red color indicated the corresponding revisions in the re-submitted files.

Comments 1: It is advisable to expand the introduction.

Responses 1: We appreciate your comments, and as per your suggestion, the introduction has been expanded accordingly. The revision can be seen in the revised manuscript, Line 59-67.

Comments 2: It is desirable to more clearly define the purpose of this study.

Responses 2: Thank you for your suggestion. We have emphasized the purpose of our study in the abstract, which can be found in the revised manuscript, Line 32-33.

Comments 3: It is advisable to compare all experimental results in more detail with data from the literature.

Responses 3: The additional discussion comparing our experimental results with other literature can be found in the revised manuscript, Line 380-385 and Line 406-407.

Comments 4: Was the experimental methodology developed by the authors or modified? What did the authors rely on in selecting the study parameters?

Responses 4: The determination of experimental design, methods, and selection of research parameters were based on our previous studies (Yang et al., 2020; Ding et al. 2021) and references from other literature (Godoy et al., 2012; Guo et al., 2022; Zheng et al., 2012). The selected research parameters include the variations of microalgae, nutrient factors, PA and FL bacterial communities, as well as the relationships among them, serving as the subjects for study and discussion. These findings enable us to achieve our goal of studying the distinct effects of various environmental factors on PA and FL bacterial communities.

Yang et al. Manipulating the phytoplankton community has the potential to create a stable bacterioplankton community in a shrimp rearing environment. Aquaculture, 2020, 520: 734789.

Ding et al. Response of the rearing water bacterial community to the beneficial microalga Nannochloropsis oculata cocultured with Pacific white shrimp (Litopenaeus vannamei). Aquaculture, 2021, 542: 736895.

Godoy et al. Effect of diatom supplementation during the nursery rearing of Litopenaeus vannamei (Boone, 1931) in a heterotrophic culture system. Aquaculture International, 2012, 20: 559-569.

Zheng et al. Bacterial community associated with healthy and diseased Pacific white shrimp (Litopenaeus vannamei) larvae and rearing water across different growth stages. Frontiers in microbiology, 2017, 8: 1362.

Guo et al. Sucrose addition directionally enhances bacterial community convergence and network stability of the shrimp culture system. npj Biofilms and Microbiomes, 2022, 8(1): 22.

Comments 5: It is advisable to expand the conclusions.

Responses 5: Thank you for your assistance. The conclusion has been expanded as per your suggestion, and the revised section can be found in the manuscript, Line 430-447.